# Multiwalled Carbon Nanotubes Embedded in a Polymeric Matrix as a New Material for Thin Film Microextraction (TFME) in Organic Pollutant Monitoring

**DOI:** 10.3390/polym15020314

**Published:** 2023-01-07

**Authors:** Ivonne Quintanilla, Carlos Perelló, Francesca Merlo, Antonella Profumo, Clàudia Fontàs, Enriqueta Anticó

**Affiliations:** 1Department of Chemistry, Universitat de Girona, C/Maria Aurèlia Capmany 69, 17003 Girona, Spain; 2Department of Chemistry, University of Pavia, Via Taramelli 12, 27100 Pavia, Italy

**Keywords:** carbon nanotubes, organic pollutants, functionalised films, TFME

## Abstract

It is essential to monitor organic pollutants to control contamination levels in environmental water bodies. In this respect, the development of new materials based on functionalised polymeric films for the measurement of toxic compounds is of interest. In this study, we prepare new films based on polymer cellulose triacetate modified with multi-walled carbon nanotubes for the monitoring of selected compounds: a fungicide (chlorpyrifos) and two emerging contaminants, the musk tonalide and the bactericide triclosan, which are used in the formulation of personal care products. The films, upon contact with water samples and following the principles of thin film microextraction, allow the determination of organic pollutants at low concentration levels. The contact time of the film with a predetermined volume of water is fixed at 60 min, and the compounds are eluted with a small volume (1 mL) of organic solvent for GC-MS analysis. Parameters such as repeatability for different films and detection limits are found to be satisfactory. Applying the method to river water demonstrates its suitability and, in the cases of chlorpyrifos and tonalide, the absence of a significant matrix effect.

## 1. Introduction

Sample treatment is an important step in the development of analytical methods, mainly to simplify the matrix in the case of especially complex matrices and to pre-concentrate an analyte when its concentration in the sample is very low. Among the different sample treatment techniques, the use of organic solvents for the extraction of organic compounds has been successfully used for many years. However, traditional solvent extraction techniques present some drawbacks related to the use of organic solvents that may be toxic, flammable, and environmentally unfriendly, and to the high amount of waste that is generated through solvent extraction [1].

To improve the performance of sample treatment, efficient processes such as solid phase extraction and microextraction techniques are being sought for the detection of organic pollutants, particularly in the case of aqueous samples. The outstanding features of microextraction techniques rely, along with their simplicity, on the small volume of the extracting phase, where the target compounds are preconcentrated. The range of applications is broad, comprising volatile and non-volatile organic compounds, with different polarities, and also ionic and metallic species [2,3,4,5].

The term “microextraction” includes different techniques depending on the extracting phase; either solid or liquid (liquid phase microextraction, LPME) and configurations: SPME, µ-SPE, single-drop microextraction (SDME), dispersive liquid-liquid microextraction (DLLME), hollow fibre liquid-phase microextraction (HF-LPME) [3,4,5] and, more recently, a new methodology called thin-film microextraction (TFME). This last technique involves the use of polymeric materials as a thin film [6,7], solving most of the limitations attributed to SPME and LPME. In relation to LPME, no loss of liquid film takes place with thin films, and compared with SPME, the greater surface area of the film avoids saturation problems. The TFME process is based on direct contact of the film with a sample for the extraction of the analytes, and subsequently, the elution of the analytes entrapped in the film using a small volume of the appropriate solvent. Various sampling formats were studied for the TFME technique. In the simplest approach, the film is placed in the sample matrix without being held in position or coated on the surface of the sample vial [5]. Alternatively, TFME keeps the film in a fixed position using a stainless-steel rod as the support, which is attached to the cap of the vial by piercing through the septum or a Teflon holder. Among the polymers, polydimethylsiloxane (PDMS) has been mainly used for this purpose [5], as well as ethylene-vinyl acetate in vial-coated TF-SPME and mixed-phase thin films of polydimethylsiloxane/divinylbenzene (PDMS/DVB) or carboxen/polydimethylsiloxane (CAR/DVB) prepared on glass wool substrates [5]. Mixed sorptive phases have been synthesized and provide balanced coverage of analytes with a broad range of chemical properties [8].

Both our group and others have previously reported the use of functionalised polymer inclusion membranes (PIMs) for the separation and preconcentration of inorganic species and antibiotics [9,10,11,12,13]. PIM preparation is simple, with the appropriate amount of the polymer and a plasticizer mixed in a desired proportion, and can be functionalised by adding extractants, typically used in liquid-liquid extraction, that act as carriers by interacting with the compound of interest. PIM stability has recently been evaluated by the addition of inorganic nanoparticles or carbon nanotubes (CNTs), which have been applied in the enrichment of arsenate and phosphate [14].

Moreover, PIMs containing a polymer and a plasticizer have been successfully used by our group in the preconcentration of organic pollutants [15,16], where the plasticizer not only provides flexibility to the film but also affords an adequate organic phase for the extraction. The application of these polymeric films was tested for the determination of pesticides (chlorpyrifos, cyprodinil, and diazinon) [15], and emerging pollutants triclosan (TCS) and tonalide (AHTN) [15]. Under exhaustive extraction conditions (contact time of six hours), absolute recoveries for pesticides of between 45–70% were obtained for the PIM-assisted extraction method and between 75–110% for AHTN and TCS. However, the leaching of the plasticizer in the aqueous solution was noticed [16], which affected the chromatographic analysis. Given this, alternative modifiers are also under investigation to improve the greenness of the methodology.

Nanomaterials, which have sizes ranging from 1 to 100 nm in one or more dimensions, are at the core of innovation in many technological areas as well as in analytical chemistry [17]. Multi-walled carbon nanotubes (MWCNTs) are composed of single-wall carbon nanotubes with additional graphene tubes around their core. The highly developed hydrophobic surface of CNTs exhibits strong sorption properties toward various compounds, and CNTs have shown great potential in adsorbing toxic ions and organic contaminants from domestic and industrial wastewater [18,19,20]. Within analytical chemistry, MWCNTs and other derived carbon materials have been used for the preconcentration of trace amounts of organic compounds from environmental samples [21,22,23,24,25,26,27] and in D-µ-SPE [28,29]. Their application in TFME is not so ubiquitous. One approach, called simultaneous liquid-liquid microextraction and CNT-reinforced hollow-fibre microporous membrane solid-liquid phase microextraction, uses a hollow-fibre porous membrane impregnated with a CNT dispersion and has been tested in the extraction of organophosphorus compounds (OP) from watermelon samples [30] and non-steroidal anti-inflammatory drugs from urine samples [31].

Due to their agricultural and domestic uses, the presence of pesticides and personal care products in river waters and effluents from wastewater treatment plants has been described worldwide [32,33]. Chlorpyrifos (CPS) is an organophosphate insecticide widely used both in agriculture and gardening. The World Health Organization considers it to be a compound of moderate toxicity for humans [34], and its use is now forbidden in Europe. CPS is included as a priority substance in the Water Framework Directive (WFD) [35], with an environmental quality standard (EQS) of 0.03 µg L^−1^. Triclosan (TCS) is commonly used as a bacteriostatic, fungistatic, mildewstat, and deodorizer. As a preservative material, TCS is used in many products, including adhesives, fabrics, vinyl, plastics, floor wax emulsions, and textiles, among others. TCS is not biodegradable and can accumulate in soil and surface water, attached to suspended sediments and affecting aquatic organisms and is regarded as an emerging pollutant [36,37]. Tonalide (AHTN) is a component used in non-alcoholic perfumes. It is the most produced synthetic polycyclic musk and can be found in a large variety of consumer products, including cosmetics, detergents, toiletries, and cigarettes. AHTN is not properly eliminated in conventional wastewater treatment plants (WWTPs) and enters rivers and bioaccumulates in the same way as TCS [38,39]. It has been demonstrated that absorption by MWCNT is an efficient technique for the removal of these compounds [29,40,41].

In this study, we investigate for the first time the use of MWCNTs embedded in a polymeric film to perform TFME of three organic pollutants, namely CPS, TCS, and AHTN. The behaviour of a 10% MWCNT-containing film is compared with a film prepared with 30% dibutyl sebacate (DBS) as the plasticizer. Our aim was to explore whether the use of the plasticizer can be avoided in the preparation of the film, thereby improving the chromatographic analysis and the green characteristics of the TFME method.

## 2. Materials and Methods

### 2.1. Reagents and Solutions

The target organic compounds were Chlorpyrifos from PESTANAL^®^ (99.8%, HPLC), Tonalide (≥98%), and Triclosan, certified reference material, from TraceCERT^®^, Sigma Aldrich (Steinheim, Germany). In Table 1, the most significant properties of the compounds are shown.

Stock standard solutions (452 mg L^−1^ for CPS, 510 mg L^−1^ for AHTN, and 1017 mg L^−1^ for TCS) were prepared in ethyl acetate. From the stock solutions, intermediate solutions at two concentration levels were also prepared in ethyl acetate and replaced every two weeks. These solutions were kept at 4 °C. Standard solutions for the calibration curve used in the calculation of absolute recoveries were made with ethyl acetate as the solvent.

Samples for TFME were prepared daily in 0.01 M NaCl in a range of 0.05 mg L^−1^ to 0.4 mg L^−1^ for CPS and AHTN and 0.2 to 4.0 mg L^−1^ for TCS.

For the preparation of the films, the following reagents were used: cellulose triacetate (CTA) from Acros Organics (Geel, Belgium) as the polymer; dibutyl sebacate (DBS) (≥97%, GC) and nitrophenyl octyl ether (NPOE) from Sigma Aldrich (Steinheim, Germany) as the plasticizers. Multi-walled carbon nanotubes (MWCNT), as produced cathode deposit, O.D. × L 7–15 nm × 0.5–10 µm, were from Sigma Aldrich (Steinheim, Germany). The XRD spectrum is shown in Appendix A. According to the producer, the C content is >99% of the TGA results.

The solvents used were chloroform (CHCl_3_) (≥99.8%), Sigma Aldrich (Steinheim, Germany); ethanol (≥99.8%, HPLC) PanReac Applichem (Castellar del Vallès, Spain); ethyl acetate (HPLC; Fisher Scientific, Madrid, Spain) and acetone (HPLC grade) by PanReac Applichem (Castellar del Vallès, Spain).

All other reagents were of analytical grade. Ultrapure water was obtained from a purified system Milli-Q plus System Millipore Iberica, S.A. (Barcelona, Spain).

### 2.2. Preparation of the Films

The films (Table 2) were prepared using the solvent casting method, with CTA as the polymer, and MWCNs as the modifier. Additionally, films containing only plasticizer (DBS) and mixtures of MWCNs/plasticizer (NPOE) were prepared for comparison. The procedure was as follows: 120 mg of CTA was dissolved in 15 mL of chloroform for 4 h; then, the desired amount of modifier was added and further agitated for 1 h. After this, the solution was poured into a Petri dish (7 cm diameter), which was set horizontally and covered loosely. The solvent was allowed to evaporate over 24 h at room temperature, and the resulting film was then carefully peeled off the bottom. In the case of films with MWCNTs, a suspension was prepared by using the appropriate number of solids (see Table 2) in 5 mL ethanol. The suspension was homogenized in an ultrasonic bath for at least 15 min and finally added to the polymer solution.

The homogeneity of the membranes was checked visually. Furthermore, the films were characterized using different techniques: infrared spectroscopy (FT-IR), scanning electronic microscopy (SEM), and transmission electronic microscopy (TEM).

FT-IR spectra were obtained with the aid of a diamond attenuated total reflectance accessory on an Agilent Cary 630 FTIR spectrometer (Agilent Technologies, Santa Clara, CA, USA). For each sample, 32 scans with a resolution of 8 cm^−1^ were recorded.

SEM images were obtained with a field emission scanning electron microscope (Hi-tachi, S-4100, Tokyo, Japan). For the surface and cross-section breakage observation, the sample preparation was as follows: a piece of the sample was cut with scissors to see both sides; another piece, around 2 cm long, was broken by cryosection and cooled in liquid nitrogen. Samples were placed on a stub and coated with carbon (model K950 turbo evaporator, Emitech, Lohmar, Germany). Digital images were collected and processed by the Quartz PCI program, version 5. The surface of the resin block prepared for TEM studies, from which the ultra-thin cuts were obtained, was also measured.

TEM studies were performed using a Jeol 1010 (Jeol, Tokyo, Japan) coupled to an Orius CCD camera (Gatan, Pleasanton, CA, USA) on the TEM-SEM Electron Microscopy Unit from the Scientific and Technological Centres of the University of Barcelona (CCiTUB, Barcelona, Spain). The samples were embedded with fresh 100% resin in silicone moulds. After 72 h polymerization at 60 °C, ultra-thin sections, of 60–80 nm thickness, were cut using an RMC PowerTome model XL ultramicrotome with a diamond knife (Diatome AG, Biel, Switzerland) and collected in 200 Mesh copper grids.

### 2.3. Thin Film Microextraction Procedure

Following the microextraction approach, the procedure was performed on 10 mL of 0.01 M NaCl solution spiked with the organic pollutants at 200 µg L^−1^ for CPS and AHTN and 2000 µg L^−1^ for TCS. A piece of the film (1 cm × 2 cm) was placed in contact with the solution for 1 h under stirring. A stainless-steel rod attached to the cap of the vial was used as the support. The film was eluted in an ultrasound bath for 15 min with 1 mL ethyl acetate. The results are given as the normalized peak area (percentages), with area values taken from the chromatogram obtained in the GC-MS upon injection of the solution after the elution step.

Once the experimental conditions had been established, the TFME procedure was then applied to standard solutions at different analyte concentrations and spiked river water for validation and application purposes.

### 2.4. Chromatographic Analysis

For the chromatographic separation, a GC-MS (DSQ, Thermo Scientific, Waltham, MA, USA) was used, entailing a TG-5SILMS capillary column (Thermo Scientific, 30 m × 0.25 mm i.d., film thickness 0.25 μm). The carrier gas was 99.9990% pure helium at a constant flow rate of 1 mL min^−1^. The split/splitless injection port was operated in splitless mode at 250 °C and maintained for 1 min. The sample injection volume was 1 μL. The thermal program was started at 60 °C, maintained for 2 min, ramped up to 150 °C at 25 °C min^−1^ and then up to 250 at 12 °C min^−1^, and held for 4 min. Ionization was performed in the electron impact mode at 70 eV. The transfer line temperature was set at 250 °C and the ion source temperature at 225 °C. The chromatographic data were acquired with Xcalibur 1.4 software (Thermo Scientific) using the SIM mode. In Table 3, the retention time (Rt) and the fragments (*m*/*z*) used for compound identification and quantitation are presented.

### 2.5. Application Studies: River Samples

To evaluate the performance of the method in real water matrices, samples from two rivers in Catalonia, namely the Llémena River (conductivity 603 μS cm^−1^, pH 8.49, total organic carbon (TOC) 1.57 mg C L^−1^) and Osor River (conductivity 247 μS cm^−1^, pH 8.58, TOC 2.03 mg C L^−1^). The samples were stored in a deep freezer at −18 °C until use. Before testing, the samples were spiked with 0.20 mg L^−1^ for CPS and AHTN and 1.26 mg L^−1^ for TCS.

The results obtained after the application of the TFME method were analysed (i) by plotting peak areas from the Osor and Llémena rivers and comparing them with the peak areas obtained with an 0.01 M NaCl matrix used throughout the work and (ii) by calculating absolute recoveries (AR) following the guideline recommendations [42].

## 3. Results

### 3.1. Preparation and Characterization of the Films

In previous works, films containing CTA as the polymer and different types of plasticizers were prepared by solvent casting and were tested for the extraction and preconcentration of organic pollutants under exhaustive extraction conditions [15,16]. A mixture of 70% of CTA and 30% of plasticizer gave flexible and homogenous films, which were transparent and mechanically stable, as F1 in Figure 1a. Using the same solvent casting procedure, films using CTA and 10% MWCNTs were satisfactorily prepared (F3, Figure 1b). The slow evaporation of the solvent under mechanical agitation (in an orbital stirrer) was crucial to prevent the aggregation of MWCNTs and to obtain a quasi-homogenous film, but less flexible compared with F1. A film containing a mixture of plasticizer and MWCNTs was also prepared (F2 in Table 2) to evaluate the effect of the plasticizer on the homogeneity of the film and the extraction performance. No differences were observed in comparing F2 and F3 with regard to the preparation behaviour and the characteristics of the film as observed by the naked eye. The F2 film, containing a mixture of both plasticizer and MWCNTs, showed intermediate mechanical behaviour.

As for the FT-IR spectra, the films with MWCNTs presented the characteristic bands of the polymer, as can be seen in Figure 2, particularly at 1734 and 1037 cm^−1^, corresponding to the C=O and C-O-C stretching vibrations in the CTA polymer. The wavenumber of the transmission band did not shift due to the presence of the carbon nanomaterials (Figure 2b).

SEM images of the F3 film are shown below (Figure 3). As can be observed, the surface of the film (Figure 3a) and the cross-section obtained after cryosection (Figure 3b) present a certain rugosity due to the presence of the nanotubes. Moreover, in the cross-section detail (Figure 3c–f), the presence of the nanotubes in the bulk of the film is even more evident. The distribution of the nanotubes in the form of layers at all depths of the membrane, rather than exclusively at the surface, can also be seen.

The TEM images (Figure 4) confirm the “layer like” distribution of MWCNTs within the bulk polymer, as was seen by SEM. Detail of the MWCNTs can also be observed in Figure 4b.

### 3.2. Interaction of Films with Embedded MWCNTs with the Organic Pollutants

The higher adsorption capacity of CNTs towards organic micropollutants, especially aromatic compounds, has been attributed to π-π interaction between the hexagonal arrays of carbon atoms in CNTs and the benzene ring. Their application in sample treatment has been demonstrated in different publications [22,23,24]. However, few examples can be found using CNTs as modifiers in polymeric coatings in TFME. Here, we have investigated for the first time new polymeric films containing MWCNTs as an alternative to the polymeric membranes prepared with plasticizers for microextraction. Preliminary tests were conducted with films containing 5% and 10% MWCNTs. Under exhaustive extraction conditions and a 6 h contact time (see details in Appendix A), with the film with 5% MWCNTs a quantitative extraction for CPS was achieved and 74% extraction for AHTN (TCS was not tested), whereas for the film with 10% MWCNTs, quantitative extraction for both compounds was obtained. In accordance with these results, the amount of MWCNTs in the film was fixed at 14 mg for membranes prepared with 120 mg of CTA (F3 film in Table 2). The same amount of MWCNTs was used in the preparation of F2 film (equivalent to 8% *w*/*w* MWCNTs), where a certain amount of plasticizer was also added to evaluate whether there was any improvement in the preparation and the extraction results. We also confirmed that there was no extraction when films prepared with 100% CTA were used.

### 3.3. Effect of Film Composition on TFME

Films F2 and F3 were tested in TFME and compared with F1. As can be observed in Figure 5, with the F3 film, an increase in the chromatographic response was produced compared with the F1 film, for CPS and AHTN microextraction. Moreover, the F2 film gave similar results to F3, showing that the plasticizer does not have a significant influence.

For TCS, the opposite behaviour was found: F1 was more favourable for the microextraction of this pollutant. TCS presents a lower log K_ow_ value (see Table 1), which makes this compound less prone to interact with the MWCNT-modified film.

Considering the results for the three compounds as a whole, the films with 10% MWCNTs seem to be a promising choice in TFME to avoid the use of plasticizers. Driven by these considerations, this formulation was used in the following experiments.

### 3.4. Elution Conditions

Ethyl acetate was selected as the eluent in the work of Merlo et al. [16] dealing with TF-LPME for the same compounds. Considering these previous studies, elution with ethyl acetate (1 mL) under ultrasound-assisted (US) elution or orbital stirring (OA) was investigated for the new films prepared in the present work. The elution time was fixed at 15 min. As can be observed in Figure 6, the response obtained depends on the compound: ultrasound-assisted elution gives better efficiency for CPS and AHTN. On the other hand, slightly better results were obtained for TCS with the aid of orbital stirring. Taking into consideration these results, we selected US elution as the final method.

### 3.5. Repeatability

The dependence of the results on the film preparation may be a factor limiting the repeatability of the method. For this, the behaviours of three films with each composition were compared by performing three replicate microextraction experiments with each film (denoted as R1, R2 and R3 in Figure 7). From these experiments, the inter-batch RSDs and intra-bath RSDs were calculated.

The inter-batch RSDs were 26% (*n* = 9), 39% (*n* = 7) and 22% (*n* = 9), for CPS, TCS, and AHTN, respectively, whereas the intra-batch RSDs (*n* = 3) were 18%, 10%, and 11% for CPS; 21% and 14% for TCS; and 9%, 11%, and 14% for AHTN. All analytes exhibited satisfactory intra-batch repeatability (RSD ≤ 20%) in accordance with validation guidelines [42]. However, better inter-batch repeatability could be achieved by improving the film preparation. This issue is currently under investigation in our group.

### 3.6. Analytical Figures of Merit

Applying the final conditions for TFME, calibration curves were constructed using an 0.01 M NaCl matrix spiked with analytes at concentrations of 50–200 µg L^−1^ for CPS and TCS and 500–4000 µg L^−1^ for TCS. The analytical figures of merit are shown in Table 4.

The linearity of the calibration curves was verified using an analysis of variance (ANOVA). All the calibration curves exhibited small *p*-values, indicating a strong correlation between the independent and dependent variables. LODs were calculated using a signal-to-noise (S/N) equal to 3 criteria. The LODs obtained, although 10 times higher, are not in disagreement with those reported in an earlier report by our group [15] considering that, in the previous work, exhaustive extraction conditions (6 h), a larger volume of the water sample, and a more sensitive instrument were employed. The main advantage here is that shorter extraction times are required, and there is less contamination due to the leaching of the plasticizer. The values found are promising and could be further lowered by modifying some of the experimental conditions, in particular by increasing the film area and reducing the volume of the eluent.

A chromatogram at the lowest concentration from the working range studied is shown in Figure 8. The compounds were well identified, with negligible noise. It is worth mentioning the pronounced tail of the chromatographic peak for TCS, which makes the integration more difficult with greater variability of the results. This is also evident from the lowest determination coefficient in Table 4.

### 3.7. Application to Water Samples

The developed method was applied in the extraction of analytes from two rivers in Catalonia, northern Spain: the Osor River (a mountain river, the sampling point was in a forest area) and the Llémena River (the sampling point was close to an urban center but in an area of farms where animals are fattened). Aliquots of the sample were first tested to ensure the absence of the target analytes. Further aliquots were then spiked at 200 µg L^−1^ for CPS and AHTN, and at 2000 µg L^−1^ for TCS. In Figure 9, absolute areas were presented together with the peak areas obtained from the 0.01 M NaCl solution spiked at the same concentration level. This plot allows a qualitative evaluation of the matrix effect. As can be seen, for CPS and AHTN, the only significant differences were found in the case of Llémena water. TCS is the compound that presents the most differences due to a stronger matrix effect.

Absolute recoveries (AR) were calculated in accordance with the validation guidelines using a single-level calibration [42]. The values, set out in Table 5, can be considered as being acceptable for the Osor River sample, considering the spiked level [43]. However, for the Llémena River, AR deviated from the accepted values, which shows the greater complexity of the matrix. These results suggest that the proposed technology can be used for studies exploring the binding of organic pollutants with water matrix components

## 4. Conclusions

In this study, a polymeric matrix embedded with MWCNTs has been proposed for the first time and successfully applied as a new phase for the TFME of organic pollutants. The inherent chemical characteristics of the MWCNTs made a significant contribution to the extraction process, as demonstrated by a cross-study between membrane composition and extraction properties. As a result, the presence of 10% (*w*/*w*) of MWCNTs in the film composition ensures an appropriate phase for the adsorption of CPS, TCS and AHTN. This makes the whole TFME procedure more environmentally friendly, avoiding the use of plasticizers. Additionally, no conditioning or derivatization steps are required in the proposed methodology, minimizing waste generation, and enabling high sample throughput. It is also important to note that the proposed membrane led to a rapid TFME (1 h for the extraction and 15 min for the elution). Moreover, the microextraction method developed here provides limits of detection of 1 µg L^−1^ for CPS and AHTN and 10 µg L^−1^ for TCS. Repeatability was good, with RSD ≤ 20% when using pieces from a single film. The application at two different spiked river water samples has shown a certain matrix effect, particularly in the case of TCS.

## Figures and Tables

**Figure 1 polymers-15-00314-f001:**
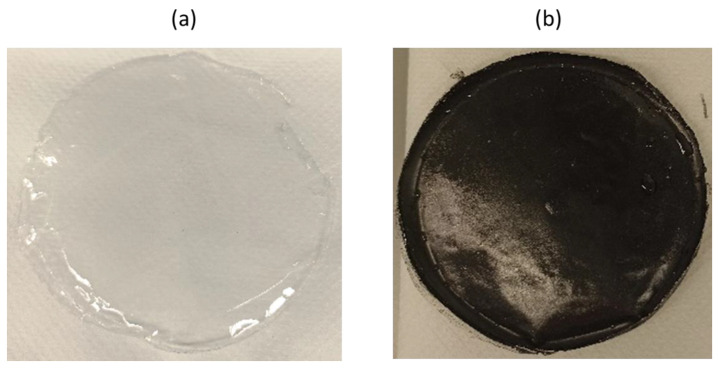
Images of (**a**) F1 (30% DBS) and (**b**) F3 (10%MWCNTs).

**Figure 2 polymers-15-00314-f002:**
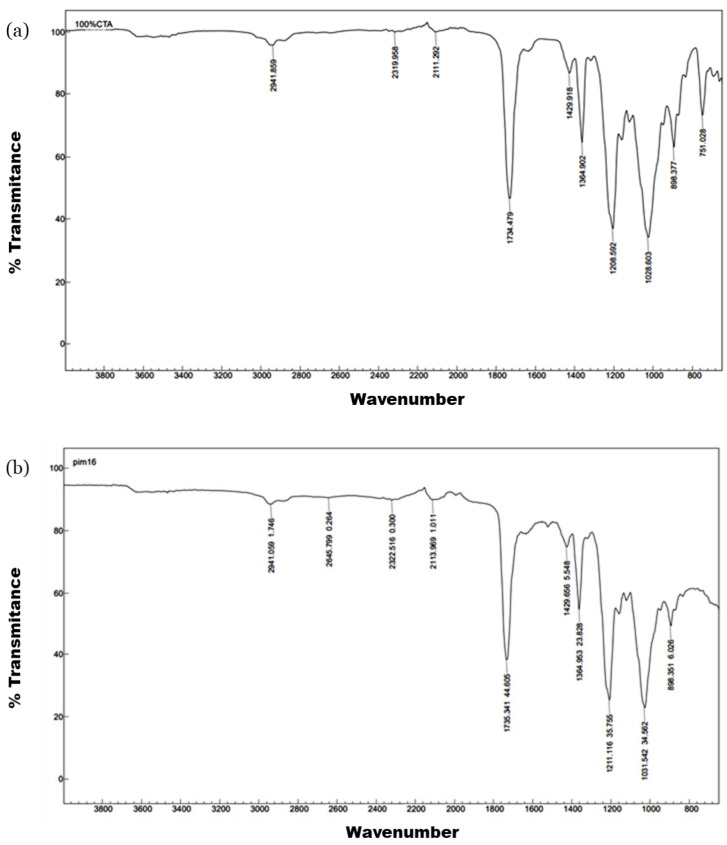
FTIR spectra comparison of (**a**) a film made of 100% CTA and (**b**) F3 (10% MWCNTs).

**Figure 3 polymers-15-00314-f003:**
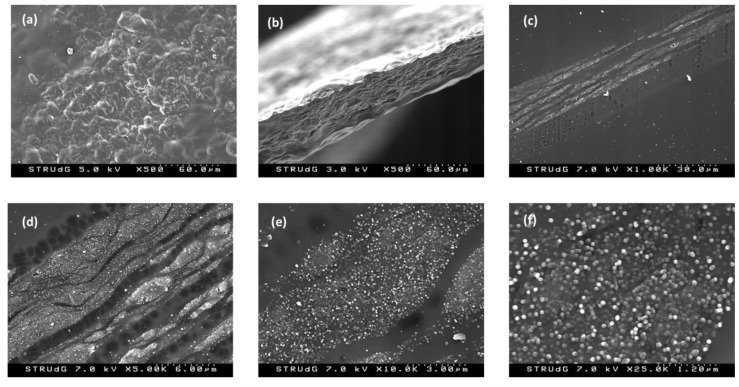
SEM of films manufactured with 10% MWCNTs, F3: (**a**) surface image; (**b**–**f**) cross sections of the film at different magnifications.

**Figure 4 polymers-15-00314-f004:**
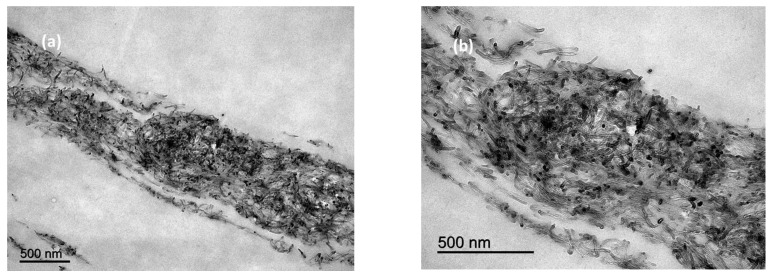
TEM images of cross sections of films manufactured with 10% MWCNTs at low (**a**,**b**) high magnification.

**Figure 5 polymers-15-00314-f005:**
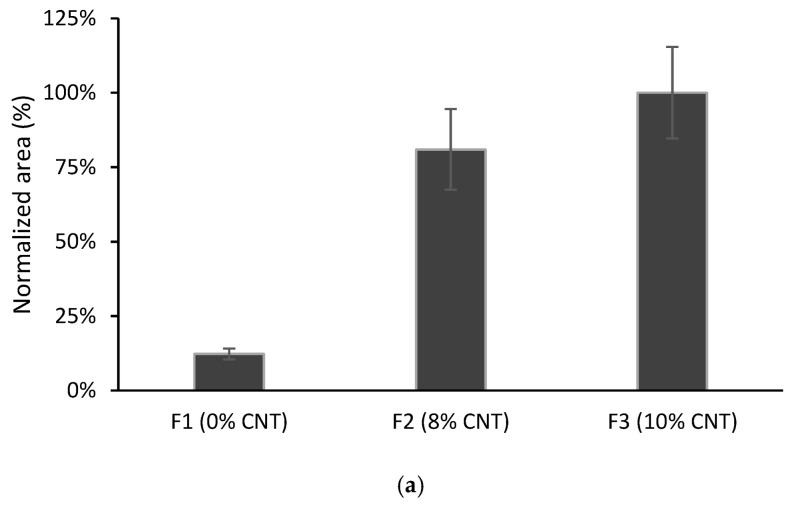
Comparison of different films containing MWCNTs and/or plasticizer: (**a**) CPS; (**b**) TCS; (**c**) AHTN. Volume of solution 10 mL spiked at 203 µg L^−1^ for CPS and AHTN and 2000 µg L^−1^ for TCS (*n* = 3).

**Figure 6 polymers-15-00314-f006:**
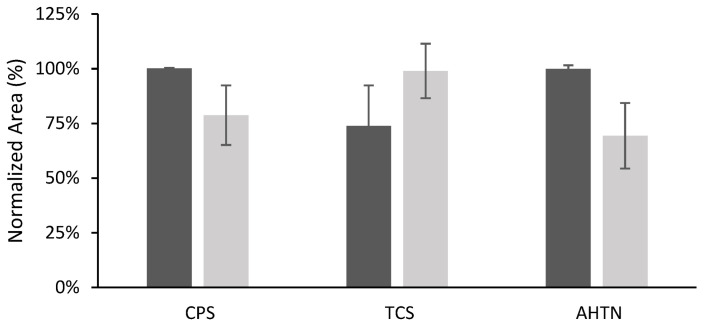
Elution results obtained using 1 mL of ethyl acetate under orbital stirring and ultrasound (*n* = 4). Volume of solution of organic compounds 10 mL spiked at 203 µg L^−1^ for CPS and AHTN and 2000 µg L^−1^ for TCS.

**Figure 7 polymers-15-00314-f007:**
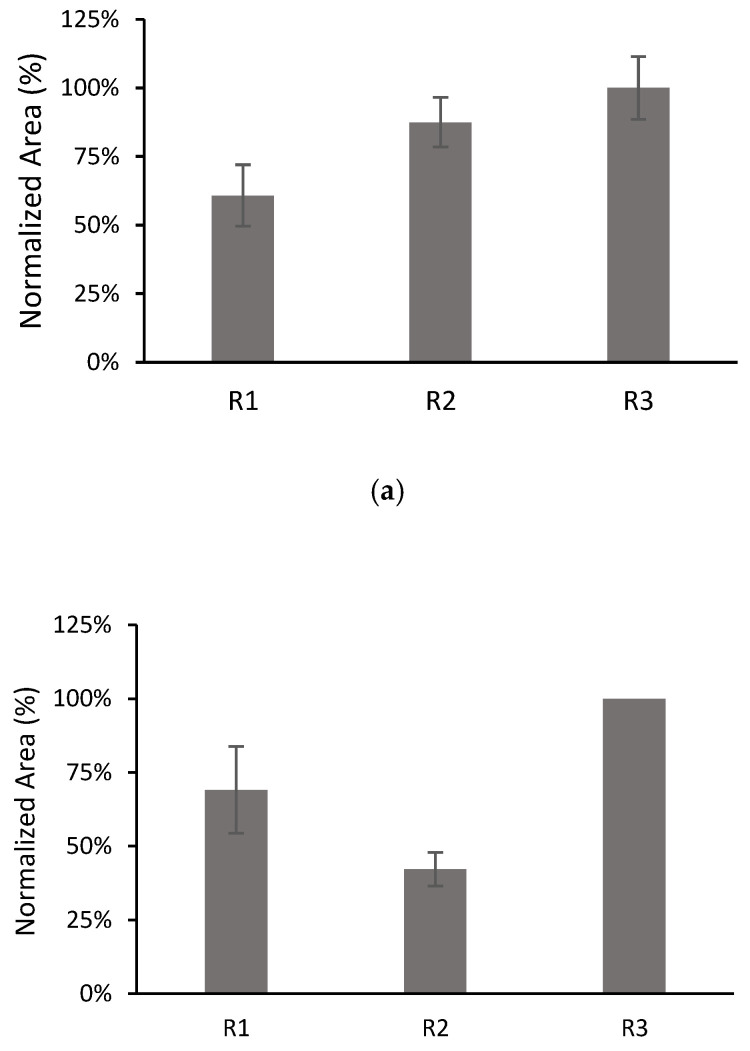
Comparison of the results for three different films with 10% MWCNTs: (**a**) CPS; (**b**) TCS; (**c**) AHTN. Volume of solution 10 mL spiked at 203 µg L^−1^ for CPS and AHTN and 2000 µg L^−1^ for TCS. The number of replicates for each film was 3, except for TCS R3 (*n* = 1).

**Figure 8 polymers-15-00314-f008:**
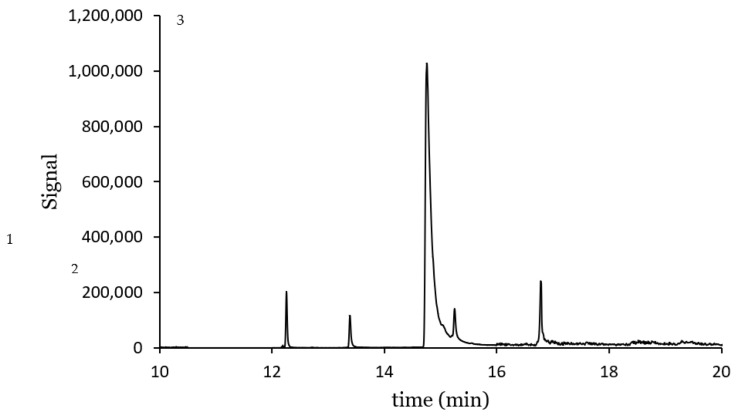
Ion extracted chromatogram for a solution containing CPS: 50 µg L^−1^, TCS: 500 µg L^−1^ and AHTN 50 µg L^−1^: 1. AHTN, 2. CPS. 3. TCS.

**Figure 9 polymers-15-00314-f009:**
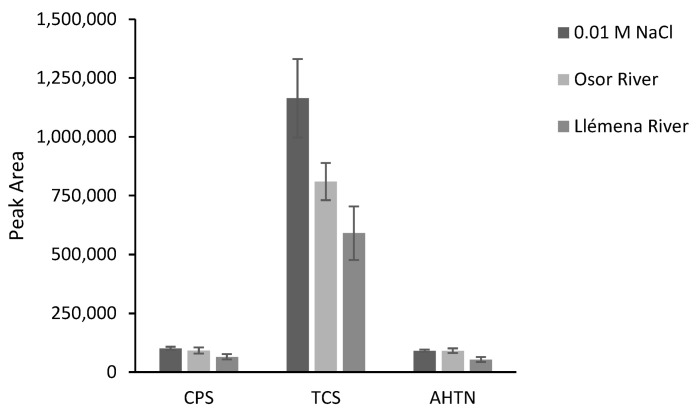
TFME response of F1 for different water matrices (*n* = 3). The samples were spiked at 200 µg L^−1^ for CPS and AHTN, and at 2000 µg L^−1^ for TCS.

**Table 1 polymers-15-00314-t001:** Properties of studied compounds.

Compound Name	IUPAC Name	Molecular Weight (g mol^−1^)	Classification	CAS Number	Log K_ow_ *
Chlorpyrifos (CPS)	O,O-Diethyl O-(3,5,6- trichloropyridin-2-yl) phosphorothioate	350.6	Organophosphorus insecticide	2921-88-2	4.96
Triclosan (TCS)	5-Chloro-2-(2,4- dichlorophenoxy)phenol	289.5	Antibacterial and fungicide	3380-34-5	4.76
Tonalide (AHTN)	6-Acetyl-1,1,2,4,4,7- hexamethyltetralin	258.398	Aroma musk	21145-77-7	5.70

* K_ow_ denotes the octanol-water partition coefficient.

**Table 2 polymers-15-00314-t002:** Film composition given in weight percentage. The amount of the component used is shown in parentheses.

Film	% CTA	% MWCNT	% Plasticizer (NPOE or DBS)
F1	70 (120 mg)	-	30 (50 mg)
F2	72 (126 mg)	8 (14 mg)	20 (32 mg)
F3	90 (120 mg)	10 (14 mg)	-

**Table 3 polymers-15-00314-t003:** Chromatographic data of the organic pollutants studied.

Compounds	Rt	*m*/*z*
CPS	13.39	197,341
TCS	14.80	218,288,290
AHTN	12.26	243,258

**Table 4 polymers-15-00314-t004:** Quality parameters for the proposed TFME method.

	CPS	TCS	AHTN
Calibration Curve	y = 9715.6x – 138,832	y = 8869.4x + 478,841	y = 13,362x – 370,528
Determination coefficient (R^2^)	0.9991	0.9853	0.9993
Working range (µg L^−1^)	50–200	500–4068	50–200
LOD (µg L^−1^)	1	10	1
LOQ (µg L^−1^)	5	50	5

**Table 5 polymers-15-00314-t005:** Absolute recoveries for spiked Osor River and Llémena River water samples (*n* = 3). Standard deviation is shown in brackets.

	AR (%)	
	Osor River	Llémena River
**CPS**	91 (13)	65 (11)
**TCS**	70 (7)	51 (10)
**AHTN**	100 (11)	59 (11)

## Data Availability

Not applicable.

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
