# Peer review of "Multiwalled Carbon Nanotubes Embedded in a Polymeric Matrix as a New Material for Thin Film Microextraction (TFME) in Organic Pollutant Monitoring"

_polymers, 2023, doi:10.3390/polym15020314_

Round 1
Reviewer 1 Report
Quintanilla et al fabricated polymer composites from different concentration of MWCNT in polymer matrix. Then, their usefulness as organic pollutant monitoring was reported. Interesting results are reported and the paper was reviewed as a candidate for Polymers. The paper provides new insight on use of polymer composites for novel pollution monitoring applications. But some revisions are necessary before publication. Some points are –
[1] In the experimental section, please provide the physical properties of the MWCNT used in present work such as purity, elemental composition, surface area, length and diameter etc?
[2] 10 wt% is a great filler content especially for MWCNT. So, it must be needful to study its dispersion in polymer matrix by high resolution microscopy such as TEM or SEM or AFM. Present SEM studies in Figure 3 are not sufficient since MWCNT particles are not visible. Eventually, some aggregation of MWCNT could be observed.
[3] Please provide SEM, Raman and XRD of MWCNT used in this work as a supporting experiment.
[4] In section 3.6, Figure 9 is cited but it is not provided in the paper? In fact, the figure provided is 1 in this section but it should be figure 8? There is a great confusion here. Please crosscheck?
[5] Please expand conclusion. Its poor in current form. Please highlight what is addressed for readers from experimental outcome performed in this work. Why this work is important etc.
[6] English is very poor. Many phrases in the paper are difficult to understand and meaningless. So, please edit the paper from native speaker before submitting the revisions.
Good Luck for revisions!
Reviewer 2 Report
In the manuscript of polymers-2102282, the authors had demonstrated that a polymeric matrix embedded with MWCNTs can be used as a new material for TFME in organic pollutants monitoring. The presence of 10% (w/w) of MWCNTs in the film composition ensures an appropriate phase for the adsorption of CPS, TCS and AHTN, avoiding the use of plasticizer. The microextraction method developed here provided limits of detection of 1 μg L^-1 for CPS and AHTN, and 10 μg L^-1 for TCS with good repeatability of RSD ≤ 20% when using pieces from a single membrane. However, the presented research results are not completely described and can not meet the publication standard of Polymers. The following comments are provided for the authors.
1. Optimization process of MWCNTs content in the composite film are not reseasonable. The authors may offer the results of composite film with lower or higher content of MWCNTs.
2. In the Figure 2., the FTIR spectrum of F3 (10%MWCNTs) are similar to the sample of F1 (30% DBS), the result is hard to understand.
3. In organic pollutants monitoring, the previous results should be compared, especially the research of Talanta 185 (2018) 291–298, to highlight the innovation of this manuscript in further.
4. To discuss the interaction of polymer materials and MWCNTs in organic pollutants monitoring.
Round 2
Reviewer 1 Report
Minor revision requested -
[1] The resolution of TEM is 500 nm in both images. However, authors claim them as different resolution. Please cross-check.
[2] In Figure 2, please make the font size of x-axis and y-axis of both images bold and clear.
[3] What is the physical mechanism for use of MWCNT based membrane for pollution monitoring?
Author Response
[1] The resolution of TEM is 500 nm in both images. However, authors claim them as different resolution. Please cross-check.
We agree with the comment that in TEM images the resolution is 500 nm both. But the length of the black bar on the down left corner is not the same. That means that for the image on the right side, the MWCNTs can be appreciated more in detail.
Accordingly, we have deleted the words “at different magnifications” in the figure caption (Figure 4).
[2] In Figure 2, please make the font size of x-axis and y-axis of both images bold and clear.
The quality of the figure has been improved and the axis and the font size of the axis title are now clearer.
[3] What is the physical mechanism for use of MWCNT based membrane for pollution monitoring?
The laboratory set up consists of placing a piece of the membrane embedded with MWCNTs in contact with the solution using a stainless-steel rod as support (see section 2.3 and graphical abstract). This is a common configuration for thin film microextraction techniques (see references 3-7). Films made of polydimethyl siloxane (PDMS) are frequently employed as sorptive phase in TFME. However, PDMS has some limitations, mainly the low affinity for polar compounds. This is the reason that new phases are continuously investigated. In our case, we exploited films made of multiwalled carbon nanotubes embedded in a polymeric matrix.
In this way, the nanotubes, distributed in the surface and in the form of layers at all depths of the membrane, can interact with analytes through π stacking, as discussed in sections 3.1, 3.2, 3.3.
Moreover, this set up could be easily fit for in situ pollution monitoring, placing the membrane directly in the matrix.
Reviewer 2 Report
The authors have addressed the comments from the reviewers, so I recommend it to be accepted for publication in Polymers.
Author Response
No modifications requested.